## [Peer Review File · The EMBO Journal]

Cellular quiescence uncouples the proteome from the transcriptome in neural stem cells

Alice Rossi, Antoine Coum, Manon Madelenat, Lachlan Harris, Stephanie Strohbuecker, Andrea Chai, Hania Fiaz, Rita Chaouni, Peter Faull, Neve Costello Haeven, William Grey, Dominique Bonnet, Fursham Hamid, Eugene Makeyev, Bram Snijders, Gavin Kelly, François Guillemot-RETIRED-, and Rita Sousa-Nunes

Corresponding author(s): Rita Sousa-Nunes (rita.sousa-nunes@kcl.ac.uk)

Review Timeline:

Submission Date:	28th Mar 25
Editorial Decision:	20th May 25
Revision Received:	15th Jul 25
Editorial Decision:	1st Oct 25
Revision Received:	29th Oct 25
Accepted:	25th Nov 25

Editor: Cornelius Schneider

Transaction Report:

Please note that the manuscript was transferred from another journal where it was originally reviewed.

Dear Editor and Referees,

We are transferring this manuscript from the journal, where – despite positive feedback and one round of revision – our interactions with the editor have reached an impasse. Specifically, Reviewer 1 had no further comments and recommended publication, and Reviewer 3 raised relatively minor points. Reviewer 2, however, was not happy with the revised manuscript, leading to a formal rejection.

Notwithstanding, we can address the remaining concerns by Reviewers 2 and 3. As explained below, we provide the key updates requested – including improved images of active, quiescent, and reactivating *Drosophila* NSCs; ANOVA analyses of multiple comparisons; and raw data requested.

Since this study uncovers a novel post-transcriptional mechanism underlying a critical cellular decision, we believe it will be of interest to the broad readership of *The EMBO Journal*.

Reviewers' comments:

Reviewer #1:

The authors have adequately addressed the reviewer's concerns. I now recommend its publication.

Reviewer #2:

This is a resubmitted manuscript that investigates stem cell quiescence in neural stem cells in *Drosophila* and mammals and suggests that during quiescence the proteome becomes uncoupled from the transcriptome through nuclear enrichment of mRNAs. It proposes that GA-rich mRNAs localize to nuclear speckles during quiescence and that nuclear retention of transcripts is a feature of quiescence. In the resubmitted manuscript, it reports "we employ *Drosophila* and mammalian neural stem cells to reveal that a mechanism for inhibiting translation in quiescence is nuclear enrichment of numerous messenger RNAs, resulting in uncoupling between transcriptome and proteome", yet there are no convincing mechanistic studies to support this statement. Although it is appreciated in the author response to reviewers: "Despite being pleased about the opportunity to submit a revised version of our manuscript to your journal last year, having performed much additional work, this was creating a very large volume of data and taking us into multiple directions", the main comments suggested by reviewers from the original submission were not addressed. It seems like instead of addressing what was needed, the authors extended the work into new areas, making it even less supported by the data provided. One of the most pressing issues was the poor image quality and use of markers to define cell types. New images were not provided and methods for quantification of fluorescence are still not adequately described and if they are, herein lies the problem. They are not suitable (measurements must account for size of nuclear and cytoplasmic compartments).

We have replaced the images of active, quiescent, and reactivating *Drosophila* NSCs, and again quantified data for this (new panel Fig. 2b) – the difference between aNSCs and qNSCs is still significant and our conclusions unchanged. We strongly dispute the negative assertions by this reviewer underlined by us, inviting a fresh assessment by *The EMBO Journal*. The first three of these assertions are broad and rather than spell out a rebuttal, we ask that the new appraisal assesses our data for support of our statements and addressing of prior concerns. Regarding fluorescence quantification, whilst it is possible to measure the total nuclear and cytoplasmic poly(A) signal in cultured cells, this is not the case for tightly packed cells in tissue. In both, the issue is compounded by the complex morphology of qNSC fibres, rendering it impossible to accurately capture the cytoplasmic volume. We thus did the next best thing, which was to measure a proxy for signal concentration in each compartment. This is a sensible approach given that the kinetics of the biochemical events underpinning cellular processes depend upon concentrations (sometimes local ones) rather than absolute amounts of reagents.

Reviewer #3:

Most of the concerns raised by the reviewers have been addressed by the authors, and they have added new experiments that have enhanced the paper. However, some important questions remain, which are as follows:

Figure 1: For multiple comparisons, a t-test is not appropriate. We recommend using ANOVA or another suitable multiple comparison test instead.

We had done pairwise t-tests comparing the samples for Dpn+ cells before and after knockdown to the control individually because considering it was an appropriate statistical approach. However, due to the reviewer's insistence, we also provide ANOVA across the various categories depicted in the figure key. The results, shown below, and all differences are significant. We can add them to the manuscript at their/the reviewer's discretion.

Figure 1d			
Samples	Measured variable	Significance	p-value
WT early WT late snx ^{2V327} snx ^{2V327/Df} snx ^{RNAi (18347GD)} snx ^{RNAi (GL00466)}	PH3- Process- NBs	****	<0.0001
	PH3+ Process- NBs	****	<0.0001
	PH3- Process+ NBs	****	<0.0001
	Dpn+ NBs	****	<0.0001
	Total detected NBs	****	<0.0001
Figure 1e			
Samples	Measured variable	Significance	p-value
48h mCherry ^{RNAi} B, 48h snx ^{RNAi} B, 48h snx ^{RNAi} R	PH3- Process- NBs	****	<0.0001
	PH3+ Process- NBs	****	<0.0001
	PH3- Process+ NBs	****	<0.0001
	Dpn+ NBs	ns	1
	Total detected NBs	****	<0.0001
72h mCherry ^{RNAi} B, 72h snx ^{RNAi} B, 72h snx ^{RNAi} R	PH3- Process- NBs	****	<0.0001
	PH3+ Process- NBs	****	<0.0001
	PH3- Process+ NBs	****	<0.0001
	Dpn+ NBs	****	<0.0001
	Total detected NBs	****	<0.0001
Figure 5d			
Samples	Measured variable	Significance	p-value
aNSC 3d-qNSC 10d-qNSC	norm log ₂ speckle/cytoplasm PSAP intensity	****	<0.0001
	norm log ₂ speckle/cytoplasm EIF3A intensity	****	<0.0001
	norm log ₂ speckle/cytoplasm MAP1B intensity	****	<0.0001

Figure 3b: While we understand the authors' explanation that absolute values were measured from regions of interest and then used to calculate the ratios, we still request that these absolute values be included or, at a minimum, disclosed to the reviewers. If the authors believe the raw values are unreliable, they should not be used in the ratio calculations, and these results should be excluded from the analysis.

We believe the reviewer refers to Fig. 2b rather than 3b. We have the raw values used to calculate the ratios for Fig. 2b and these are provided in a separate file.

Figure 5a: The authors should provide a clearer explanation of what is being calculated in this figure. If, as we understand, the authors are taking all nuclear-biased transcripts and calculating the fraction of different types (e.g., GA-rich, GC-rich, C-rich, etc.), we question why the sum of these ratios is approximately 120%. We recommend that the manuscript be carefully proofread to ensure all experiments are described with sufficient detail.

Figure 5a plots the average fraction of each coding transcript in the mouse transcriptome that is positive (i.e., **overrepresented**) for a specific multivalent category. We are aware that the sum is >1 in some cases, which is not an oversight. Because the classes of multivalency identified by the algorithm used are somewhat redundant, a stretch of RNA rich in repetitive Gs, As, and Cs could contribute to making a transcript region positive for both GA-rich and GC-rich multivalencies, depending on the k-mer window being examined.

The algorithm used was that developed by the Ule lab and described in Faraway *et al.*, 2024 (cited), from which we quote from there:

“repetitive arrangement of similar sequence motifs over long regions, termed multivalent regions [...]. To generalise the analysis of multivalent RNA sequences beyond specific motif sets, we designed a scoring algorithm that assigns a generalised RNA multivalency (GeRM) score for each nucleotide within provided RNA sequences (Figure 1A). For a given k-mer at each position along a transcript, the GeRM score is calculated by taking all k-mers in a defined window around the given k-mer. [...] This similarity is weighted based on the proximity to the given k-mer, and the weighted similarity scores are then summed for each k-mer (Figure 1A). Thus, k-mers that have a large number of similar k-mers nearby receive a high GeRM score, indicating high multivalency potential (Figure 1B).”

Figure 5d:

a. What statistical test was used for this analysis? A multiple comparison test (e.g., ANOVA) should be applied in this case.

We apologise for the oversight in omitting information on the statistical test and significance in the legend of this figure, which we have now added. We performed t-tests as our aim was to make pairwise comparisons between the active condition and each of the quiescence depths, but have now also added the results of ANOVA testing to the legend. Regardless, when such a high number of data points is being compared, simple statistical analysis like t-test and ANOVA can be misleading, which is why we added the median scores for each group to the graph, as we did for the violin plots in figure 4c (also explained further below).

b. Why do the authors claim that the negative control (Psap) shows no phenotype? The difference between active NSC and qNSC at both 3 days and 10 days is statistically significant, and the expression profile of Psap closely resembles that of the positive control, Map1b. Given this, we believe the authors should reconsider their conclusions about this experiment. If the authors have applied a specific fold-change threshold that has not been disclosed, we would appreciate knowing the threshold used, as well as the scientific justification for its biological relevance and significance.

The statistical comparisons of normalised Psap signal in the speckle/nucleoplasm between aNSC and qNSC are significant because we are comparing a huge number of data points (single cells from 2 biological replicates), which is biased to return significant p-values even when only minimal differences occur. We would like to draw the attention of the reviewer to the fact that the median of the relative speckle signal of Psap does not change much as a result of quiescence induction, while that of nuclear biased Eif3a and Map1b do, becoming higher, and thus relocalising to the speckle in quiescent NSCs. This is supported by the images in the corresponding supplementary figure (Supplementary Figure 6), which show that though the pattern of Psap staining becomes more speckle-like in quiescence, its distribution is evenly distributed between the nucleus and the cytoplasm, while that of Eif3a and Map1b becomes starkly more speckle-like exclusively in the nucleus with longer exposure to BMP4. As the p-value (whether from t-test or ANOVA) is misleading here due to the number of data points in each sample, we suggest omitting it from the figure (while keeping it in the legend) and showing the median of each sample, as was done in a similar scenario in Figure 4c, as per our statistician recommendation.

Dear Dr. Sousa-Nunes,

Thank you for submitting your manuscript for consideration by the EMBO Journal. Please find enclosed the comments by the arbitrating reviewer whom we had asked to evaluate the manuscript. As you can see from the report, this referee thinks that the manuscript is interesting and the data of high quality. We think that the additional concerns are fair and balanced and can be addressed with textual edits and more detailed analysis of the existing MS data.

Given the referees' positive recommendations, I would like to invite you to submit a revised version of the manuscript, addressing the comments of the reviewer. I should add that it is EMBO Journal policy to allow only a single round of revision, and acceptance of your manuscript will therefore depend on the completeness of your responses in this revised version.

We generally allow three months as standard revision time. As a matter of policy, competing manuscripts published during this period will not negatively impact on our assessment of the conceptual advance presented by your study. However, we request that you contact the editor as soon as possible upon publication of any related work, to discuss how to proceed.

Thank you for the opportunity to consider your work for publication. I look forward to your revision.

Yours sincerely,

Cornelius Schneider, PhD
Editor
The EMBO Journal
c.schneider@embojournal.org

We realize that it is difficult to revise to a specific deadline. In the interest of protecting the conceptual advance provided by the

work, we recommend a revision within 3 months (18th Aug 2025). Please discuss the revision progress ahead of this time with the editor if you require more time to complete the revisions. Use the link below to submit your revision:

Referee #1:

The study by Rossi et al focuses on the posttranscriptional mechanisms regulating NSC quiescence state. They present here evidence that nuclear retention of mRNAs is a feature of quiescence in both fly and mammalian systems. Interestingly, splicing is not inhibited in quiescent cells, suggesting that nuclear retention is not due to inhibition of splicing. A large proportion of nuclear-biased transcripts are GA-rich and localize to nuclear speckles, indicating a further level of compartmentalization. Overall, this work propose a novel posttranscriptional mechanism for regulation of protein synthesis in quiescent stem cells.

General comments

This is an intriguing report suggesting a novel mechanism potentially contributing to reduced translational output in quiescent neural stem cells. This work is novel and will foster new research in this exciting area. The present study has been improved through multiple rounds of review and relies on high quality data and rigorous data analysis and interpretation. I have a number of comments (mostly at the level of data discussion) that the Authors should address to improve the study even further.

Main comments

- Data in Figure 1 and 2 (and related S Figs) are compelling. Though, I am unconvinced it is worth including the GSC data. I would rather focus on non-neoplastic cells. Or at least argue why such a different model is useful to support the conclusions of the current study.
- In Figure 3 (or corresponding S Fig), it would be important to dig more with respect to the proteins, which are UP-regulated upon deeper quiescence transition, as some of those could be relevant as well. It feels like a missed opportunity at present. See for instance the data in subsequent figures on cyto vs nuclear fractions, where mRNAs enriched in cytoplasmic fractions are also discussed. I strongly advise the authors to look more in-depth at the upregulated proteins (see above) and add couple of panels, along with discussion.
- I am not fully convinced about the discussion on splicing of nuclear vs cytoplasmic mRNAs in the nuclear fraction of qNSCs. If those enriched mRNAs are generally more spliced in qNSCs, one could argue that it is not nuclear retention but rather nuclear re-import of processed mRNAs (unlikely but still worth commenting), or that nuclear retention increases rate of splicing? But then, why would splicing occur upon entry into deeper quiescence? Would it make more sense for a form of quiescence which is more readily reversible? More generally, the authors should discuss more along these lines.
- Finally, what I felt is missing in the discussion section are to aspects: i) what is the Authors' view on how nuclear retention mechanisms would be modulated upon EXIT from quiescence; ii) could mTOR signaling regulation be linked to nuclear retention mechanisms? Maybe a review of existing literature in the area would provide some ideas to be briefly covered in the discussion.
- I noticed that during the revision process, some of the data were removed on nucleoporins to be utilized in a different manuscript/story, if I understood correctly. The Authors should clarify why this is necessary/recommended. Why the removed data would not help the present manuscript?

We thank the referee for their helpful comments, all of which we have addressed in order to improve our manuscript. A point by point response follows.

Referee #1:

The study by Rossi et al focuses on the posttranscriptional mechanisms regulating NSC quiescence state. They present here evidence that nuclear retention of mRNAs is a feature of quiescence in both fly and mammalian systems. Interestingly, splicing is not inhibited in quiescent cells, suggesting that nuclear retention is not due to inhibition of splicing. A large proportion of nuclear-biased transcripts are GA-rich and localize to nuclear speckles, indicating a further level of compartmentalization. Overall, this work propose a novel posttranscriptional mechanism for regulation of protein synthesis in quiescent stem cells.

General comments:

This is an intriguing report suggesting a novel mechanism potentially contributing to reduced translational output in quiescent neural stem cells. This work is novel and will foster new research in this exciting area. The present study has been improved through multiple rounds of review and relies on high quality data and rigorous data analysis and interpretation. I have a number of comments (mostly at the level of data discussion) that the Authors should address to improve the study even further.

Main comments:

- Data in Figure 1 and 2 (and related S Figs) are compelling. Though, I am unconvinced it is worth including the GSC data. I would rather focus on non-neoplastic cells. Or at least argue why such a different model is useful to support the conclusions of the current study.

In probing and including GSC data, our aim was to check whether the fundamental finding of nuclear-biasing of polyadenylated RNA in quiescent stem cells also applied to pathological contexts. We looked at glioblastoma stem cells since they are also neural. This was a minor point and we have now removed panels D,E from Fig. EV3.

- In Figure 3 (or corresponding S Fig), it would be important to dig more with respect to the proteins, which are UP-regulated upon deeper quiescence transition, as some of those could be relevant as well. It feels like a missed opportunity at present. See for instance the data in subsequent figures on cyto vs nuclear fractions, where mRNAs enriched in cytoplasmic fractions are also discussed. I strongly advise the authors to look more in-depth at the upregulated proteins (see above) and add couple of panels, along with discussion.

We agree that this proteomics dataset will be a great resource for the investigation of molecules and processes that are upregulated in qNSCs, clearly much less known and understood than downregulated ones. Though an in-depth investigation of these is beyond the scope of this manuscript, we have now added the following:

*1) We draw attention to the main GO biological process terms also in the main text (new text in **bold**): "Biological process GO analysis revealed the most downregulated categories of proteins in qNSCs as (m)RNA processing and metabolism, as well as DNA organization and metabolism, **while the most upregulated categories pertained to oxidative phosphorylation and cell adhesion** (Fig. 3D and Table EV4)." Addition of further GO biological processes of upregulated proteins (eg., such that they would be 10 like the downregulated ones), would mean repetition (despite looking at the condensed list of terms), unhelpful generality, or poor significance, so we opted out of this.*

2) We highlighted four more upregulated proteins in panel 3C, namely Atp1b2, Aldh1a1, S100B, and ApoE which, like previously highlighted Id1, were also described as upregulated at the mRNA level in Blomfield et al. 2019:

We show below a list of the top 10 most upregulated proteins in qNSCs, for the reviewer's ease of viewing - it is an excerpt of Table EV3. Three of these (in bold) are ones we have now also highlighted in the new panel 4C. The fourth of those, *Aldh1a1*, was the 23rd most upregulated among proteins detected.

Protein	Gene description	Max log ₂ intensity change
Slc1a2	Solute Carrier Family 1 Member 2	4.07
Hspb8	Heat Shock Protein Family B Member 8	4.02
Atp1b2	ATPase Na ⁺ /K ⁺ Transporting Subunit Beta 2	3.98
S100b	S100 calcium binding protein B	3.81
Cd9	CD9 molecule	3.73
ApoE	Apolipoprotein E	3.65
Slc39a12	Solute Carrier Family 39 Member 12	3.61
Glud1	Glutamate Dehydrogenase 1	3.61
Atp1a2	ATPase Na ⁺ /K ⁺ Transporting Subunit Alpha 2	3.44
S100a4	S100 calcium binding protein A4	3.39

3) We had generated panel 4E to stress the impact of altered nucleocytoplasmic partitioning on corresponding protein expression, focussing on the correspondence of subcellular compartment biasing and protein up- or downregulation. We have now gone deeper into this by plotting the magnitude of protein level changes of the top 10 most nuclear-enriched and cytoplasmic-enriched transcripts (Z-score change in 10d-BMP4 vs 0d-BMP4 conditions), which we have included as a new panel B in Fig EV5:

This exemplifies specific transcripts that become strongly cytoplasm- or nuclear-biased in qNSCs whose encoded proteins become up- or downregulated, respectively. It shows that the directionality, more than the magnitude, of change is generally as expected. This strongly supports the notion of a functional significance of nuclear or cytoplasmic mRNA biasing, but of course we know that other regulation (eg., post-translational modifications) will be at play also to modulate protein levels. It should also be noted that an order of magnitude more transcripts can be detected by (frac-)RNAseq than proteins by proteomics, and the level of correspondence between the two datasets, where overlap existed, puts forward strong mRNA candidates for further investigation at the protein level.

We had generated panel 4G to look globally at the intersection of proteomic and transcriptomic data leading to protein downregulation. When doing the same for protein upregulation, i.e., biological process GO terms pertaining to upregulated proteins whose corresponding transcripts were cytoplasmic-enriched at both 3d- and 10d-qNSCs, the only significant term in the analysis related to actin filament organisation, likely due to the limited number of transcripts that become cytoplasmic-biased in quiescence compared to those that become nuclear-biased. Hence, not having created a new panel for this.

- I am not fully convinced about the discussion on splicing of nuclear vs cytoplasmic mRNAs in the nuclear fraction of qNSCs. If those enriched mRNAs are generally more spliced in qNSCs, one could argue that it is not nuclear retention but rather nuclear re-import of processed mRNAs (unlikely but still worth commenting), or that nuclear retention increases rate of splicing? But then, why would splicing occur upon entry into deeper quiescence? Would it make more sense for a form of quiescence which is more readily reversible? More generally, the authors should discuss more along these lines.

We appreciate this point and have now expanded the discussion to address the reviewer's comments. Re-written Discussion paragraphs are as follows:

“We found that most nuclear-biased mRNAs in qNSCs were completely spliced and encode (m)RNA processing regulators, whereas the minority with increased intron-retention mostly encode cell-cycle and mitotic regulators. We postulate that differential splicing levels between distinct transcript groups could underlie the sequential deployment of factors during quiescence entry and, presumably the reciprocal during quiescence exit, and orchestrate a finely tuned sequence of events whereby gene products relying differentially on transcription, splicing and/or cytoplasmic translocation will need more or less time for deployment. We note that this could even trigger feedforward loops where earlier factors promote the readiness of others to be expressed, thus favouring a prompt reversibility. A recent study reported widespread IR in various quiescent cell types⁵⁰ but did not report on qNSCs, nor did we find this in our study. It is reasonable to speculate that different cell types adopt distinct molecular strategies towards the same goal of selective nuclear biasing of transcripts as a general mechanism of translation repression in quiescence.

Several mechanisms could explain why most nuclear-biased transcripts are more fully spliced in quiescence (Supplementary Fig. 5c). While nuclear-biased transcripts could have been reimported into the nucleus in their fully spliced form, this is not parsimonious, especially in a cellular context characterised by low biosynthetic requirements like quiescence. Moreover, we did not find the upregulation of specific mRNA import pathways in our data, which rather supports a scenario where bidirectional transport is downregulated in quiescence. We hypothesise that nuclear-biased transcripts have lower intron retention due to their extended residence times in the nucleus, specifically at the speckle (Fig. 5b-c). It is also possible that some nuclear-biased transcripts benefit especially from the few splicing proteins that are upregulated in qNSC (Fig. 3E).

The fact that nucleocytoplasmic compartmentalization of most transcripts was not significantly altered between aNSCs...”

- Finally, what I felt is missing in the discussion section are two aspects: i) what is the Authors' view on how nuclear retention mechanisms would be modulated upon EXIT from quiescence; ii) could mTOR signalling regulation be linked to nuclear retention mechanisms? Maybe a review of existing literature in the area would provide some ideas to be briefly covered in the discussion.

*Regarding point i), we had referred to considering it likely that quiescence exit might involve a reverse sequence of steps to that of entry: “We thus postulate that sequential deployment of types of factors during quiescence entry and, presumably the reciprocal during quiescence exit, orchestrate a finely tuned sequence of events [...]” Our data (and that of others) strongly suggests feedforward loops that can drive cells from shallow into deeper quiescence, and imagine that a variety of cues may nudge cells to initiate the opposite feedforward sequences that can eventually lead to exit from quiescence. We have now added in the discussion the text in **bold** below:*

*“The observations here presented, in conjunction with recent data on GA-multivalent mRNA accumulation in nuclear speckles¹⁶, suggest that altered speckle composition in qNSCs might turn it into a “sink” for nuclear biasing of at least some mRNAs, particularly GA-rich multivalent ones (Fig. 5e). **Conversely, it is possible that upon quiescence exit the properties of the speckles change so that nuclear-biased mRNAs can be released, exported to the cytoplasm, and translated to drive reactivation. Work by the Ule lab¹⁶ has implicated SR protein phosphorylation by CDC-like***

kinases (CLKs) as mediating the composition and properties of the speckle. Future work will explore a potential role for nuclear speckles in quiescence regulation, with a particular focus on whether manipulating speckle properties is sufficient to control the transition between a proliferative and quiescent state.”

Regarding point ii), we have now added to the Discussion also the following text (in **bold**):
“We envision that the post-transcriptional mechanisms here uncovered interplay with other (post)transcriptional and (post)translational ones in a series of positive-feedback loops, whereby initially small changes in protein or cytoplasmic transcript abundance are amplified, underlying the continuum of states between active and deeply quiescent cells. **One key question is how the mechanism uncovered here is integrated with the TOR pathway towards a concerted cellular program of quiescence entry and exit. Obvious direct connections we looked into were whether TOR kinases regulate SR protein phosphorylation directly (lower TOR leading to SR hyperphosphorylation and consequent enrichment in speckles) or if TOR pathway components were to be GA-multivalency-rich. We found no evidence for these scenarios and future work will be needed to reveal at what level(s) the two pathways combine coherently.”**

- I noticed that during the revision process, some of the data were removed on nucleoporins to be utilized in a different manuscript/story, if I understood correctly. The Authors should clarify why this is necessary/recommended. Why the removed data would not help the present manuscript?

During the course of our work, we investigated the functional role of Nups in mammalian NSC quiescence regulation by assessing the impact of individual knockdowns on both the global cellular transcriptome and the altered nucleocytoplasmic partitioning of transcripts, which is the focus of the current manuscript. We found that downregulation of Nup98-96 or of Nup93 in mouse NSCs, to the extent possible without induction of cell death, resulted in the differential expression of hundreds of genes, and that Nup98 binds to the promoter of also hundreds of genes in active NSCs, but not in quiescent NSCs transcripts (1/3 of which are differentially expressed between qNSCs and aNSCs). Nonetheless, these knockdowns did not lead to significant differential partitioning of transcripts in mouse NSCs. We thus concluded that the impact of moderate downregulation of individual Nups on nuclear biasing of mRNA in quiescence is indirect in murine NSCs. The large body of data on the role of Nups as transcriptional regulators of NSC quiescence would deviate from our main message that interstasis contributes to the novel observation of altered nucleocytoplasmic partitioning of numerous transcripts in qNSCs. For this reason, as well as due to the amount of data acquired on that non-canonical role of Nups in regulation NSC quiescence, we decided to collate all Nup data on mouse NSCs in a different manuscript.

Dear Dr. Sousa-Nunes,

Thank you for submitting a revised version of your manuscript. There remain only a few mainly editorial points that have to be addressed before I can extend formal acceptance of the manuscript:

- Please double-check to make sure to all relevant funding information in the manuscript is also entered into our submission system.
- On the abstract page of the manuscript, please include 4-5 general keyword terms to enhance searchability.
- Please rename the Conflict of Interest section into "Disclosure and Competing Interests Statement", in accordance with our updated Guide to Authors (<https://www.embopress.org/competing-interests>)
- As we are switching from a free-text author contribution statement towards a more formal statement based on Contributor Role Taxonomy (CRediT) terms, please remove the present Author Contribution section and instead specify each author's contribution(s) directly in the Author Information page of our submission system during upload of the final manuscript. See <https://casrai.org/credit/> for more information.
- DATASET EV LEGENDS: source file names, titles, legends and manuscript callouts all need to be updated to Dataset EV1-EV# instead of Tables EV3-EV8, legends should be removed from ms and uploaded as a separate tab/sheet in each Excel file
- Please make sure that the aspect ratio of the synopsis image conforms to our website's format - it should be exactly 550 pixels wide and between 300-600 pixels high.
- In Figure EV7 B&C the Bottom cells are reused (10d-qNSC) and are different from the source data. Could you please recheck the data and update as necessary. An explanation would be appreciated.
- The source data of figure EV5A show a pattern that might indicate that not always the right columns might have been used for the figure (please find the attached source data file). Could you please doublecheck and explain.
- Figure Legends (main + EV):
 1. Please note that the exact p values are not provided in the legends of figures 1D, E; 2B, D, F, G; 5C; EV1 E, EV3 C
 2. Please indicate the statistical test used for data analysis in the legends of figures 4D, G.
 3. Please note that the error bars are not defined in the legends of figures 5C, D.
 4. Please note that the scale bar needs to be defined for figure 5B
- "Abstract" needs to be labeled, and not bolded
- Sections need to be named and the order should be corrected: Title page - Abstract - Keywords - Introduction - Results - Discussion - Methods - Data Availability - Acknowledgements - Disclosure and Competing Interests Statement - References - Figure Legends - Table(s) - Expanded View Figure Legends.

With best regards,

Cornelius Schneider

Please refer to our figure preparation guideline in order to ensure proper formatting and readability in print as well as on screen:

See also figure legend guidelines:

<https://www.embopress.org/page/journal/14602075/authorguide#figureformat>

Use the link below to submit your revision:

Point-by-point response to editorial revisions

(requests by Dr Cornelius Schneider on 1st Oct 2025
and some subsequent information by Ms Lela Djordjevic-Ristanovic)

Dear Dr. Sousa-Nunes,

Thank you for submitting a revised version of your manuscript. There remain only a few mainly editorial points that have to be addressed before I can extend formal acceptance of the manuscript:

We thank the handling editors for the care and patience with this process, which has involved several authors who have since moved countries. We have addressed all editorial concerns raised (please see below) but remain available should any further action be required.

- Please double-check to make sure to all relevant funding information in the manuscript is also entered into our submission system.

We checked and this was complete.

- On the abstract page of the manuscript, please include 4-5 general keyword terms to enhance searchability.

We have now done so.

- Please rename the Conflict of Interest section into "Disclosure and Competing Interests Statement", in accordance with our updated Guide to Authors (<https://www.embopress.org/competing-interests>)

We have now done so.

- As we are switching from a free-text author contribution statement towards a more formal statement based on Contributor Role Taxonomy (CRediT) terms, please remove the present Author Contribution section and instead specify each author's contribution(s) directly in the Author Information page of our submission system during upload of the final manuscript. See <https://casrai.org/credit/> for more information.

Lela: "I have checked our online system and noticed that you have included the author contribution in the author list, as in the attached screenshot of your account in the Author Information. So this has been resolved, thank you!"

- DATASET EV LEGENDS: source file names, titles, legends and manuscript callouts all need to be updated to Dataset EV1-EV# instead of Tables EV3-EV8, legends should be removed from ms and uploaded as a separate tab/sheet in each Excel file

Lela: "Regarding the EV Tables/Datasets, please note that the difference between EV tables and Datasets in our system is about the size of these tables, not the content. EV tables should roughly fit onto one A4 page, so that they are correctly converted to PDF. If much larger than that, they should be made datasets, as datasets are more complex, having more rows, columns, tabs/sheets.

Also, EV tables need to be editable, and I noticed that yours are pictures inserted in Word files. The nomenclature should remain Table EV1-EV2, and the legends should be removed from manuscript file.”

We have now provided Tables 1 and 2 as XLS files with the title on the same sheet and above the table, all printable as an A4 page.

“Source file names, titles, legends and manuscript callouts all need to be updated to Dataset EV1-EV6 instead of Tables EV3-EV8, legends should be removed from manuscript file and uploaded as a separate tab/sheet in each Excel file.”

We have now done so.

- Please make sure that the aspect ratio of the synopsis image conforms to our website's format - it should be exactly 550 pixels wide and between 300-600 pixels high.

We have now done so.

- In Figure EV7 B&C the Bottom cells are reused (10d-qNSC) and are different from the source data. Could you please recheck the data and update as necessary. An explanation would be appreciated.

This was an oversight on our part, for which we apologise. The source data pictures are correct and have now been used to replace the repeated ones in the figure.

- The source data of figure EV5A show a pattern that might indicate that not always the right columns might have been used for the figure (please find the attached source data file). Could you please doublecheck and explain.

Please note that though we provided the raw data of Ct values from the thermocycler, what we plotted were the fold changes of each probe in each fraction (in the source data file these are labelled as "fold change" and can be found in the tabs labelled "probe name - condition", to the right, columns M-O). We replotted these to make sure we didn't accidentally use wrong values, which produced the same graphs as those in the figure. If any concern remains, could the editor please be a bit more precise as to which columns seem wrong?

- Figure Legends (main + EV):

1. Please note that the exact p values are not provided in the legends of figures 1D, E; 2B, D, F, G; 5C; EV1 E, EV3 C

If the p-value is lower than 0.0001, Prism only reports it as <0.0001. We have now added this information to the legends, kept the asterisk legends for p < X or Y etc, and added all exact p-values provided by Prism when p > 0.0001.

2. Please indicate the statistical test used for data analysis in the legends of figures 4D, G.

We have now done so.

3. Please note that the error bars are not defined in the legends of figures 5C, D.

We have now done so.

4. Please note that the scale bar needs to be defined for figure 5B

We have now done so.

- "Abstract" needs to be labeled, and not bolded

We have now done so.

- Sections need to be named and the order should be corrected: Title page - Abstract - Keywords - Introduction - Results - Discussion - Methods - Data Availability - Acknowledgements - Disclosure and Competing Interests Statement - References - Figure Legends - Table(s) - Expanded View Figure Legends.

We have now done so.

Dear Dr. Sousa-Nunes,

I am pleased to inform you that your manuscript has been accepted for publication in the EMBO Journal.

Yours sincerely,

Cornelius Schneider, PhD
Editor
The EMBO Journal
c.schneider@embojournal.org

Please note that it is The EMBO Journal policy for the transcript of the editorial process (containing referee reports and your response letters) to be published as an online supplement to each paper. If you should prefer removal of any referee-only figures included in the point-by-point response(s), e.g. because they may still be used for future publication or because they have been reproduced from published work by others, please do let us know immediately via response email.

More information is available here: https://www.embopress.org/transparent-process#Review_Process
